# Is Frailty Diagnosis Important in Patients with COPD? A Narrative Review of the Literature

**DOI:** 10.3390/ijerph20031678

**Published:** 2023-01-17

**Authors:** Francisco José Tarazona-Santabalbina, Elsa Naval, Juan María De la Cámara-de las Heras, Cristina Cunha-Pérez, José Viña

**Affiliations:** 1Geriatric Medicine Department, Hospital Universitario de la Ribera, Carretera de Corbera km. 1, 46600 Alzira, Spain; 2Centro de Investigación Biomédica en Red Fragilidad y Envejecimiento Saludable (CIBERFES), 28029 Madrid, Spain; 3Medical School, Universidad Católica de Valencia San Vicente Mártir, 46001 Valencia, Spain; 4Department of Pneumology, Hospital Universitari la Ribera, 46600 Alzira, Spain; 5Library Service, Hospital Universitario la Ribera, 46600 Alzira, Spain; 6School of Doctorate, Universidad Católica de Valencia San Vicente Mártir, 46001 Valencia, Spain; 7Departament of Physiology, Universitat de Valencia, 46010 Valencia, Spain

**Keywords:** COPD, frailty, comprehensive geriatric assessment, outcomes, interventions, geriatric syndromes

## Abstract

Frailty is prevalent in older adults and is related to a worsening functionality, quality of life, and health outcomes. Though there is an increasing interest in this field, the relationship between frailty and worsening COPD outcomes remains unknown. A narrative review of the literature with studies published between 2018 and 2022 was carried out to address three questions: the prevalence of frailty and other geriatric syndromes in COPD patients, the link between frailty and worsening health outcomes in COPD patients, and the non-pharmacological interventions performed in order to reverse frailty in these patients. A total of 25 articles were selected. Frailty prevalence ranged from 6% and 85.9%, depending on the COPD severity and the frailty measurement tool used. Frailty in COPD patients was related to a high prevalence of geriatric syndromes and to a high incidence of adverse events such as exacerbations, admissions, readmissions, and mortality. One study showed improvements in functionality after physical intervention. In conclusion, the prevalence of frailty is associated with a high incidence of geriatric syndromes and adverse events in COPD patients. The use of frailty screenings and a comprehensive geriatric assessment of COPD patients is advisable in order to detect associated problems and to establish individualized approaches for better outcomes.

## 1. Introduction

The generally agreed-upon definition of frailty states that frailty pertains to “a medical syndrome with multiple causes and contributors that is characterized by diminished strength, endurance, and reduced physiological function that increases an individual’s vulnerability for developing increased dependency and/or death” [1]. From a clinical perspective, frailty implies a decrease in physiological reserves and an increased vulnerability to stressors, leading to greater overall vulnerability [2]. This vulnerability increases the risk of adverse health outcomes, such as disability, hospitalizations, and mortality [2]. Frailty is dynamic in nature, and it is possible to identify positive and negative trajectories among the robust, pre-frailty, and frailty categories, both spontaneously and after interventions [3]. Despite having previously mentioned the generally accepted definition of frailty, there are a large number of operational definitions and measurement tools [4], including the phenotypic model established by Linda Fried (FFP, short for Fried frailty phenotype), one of the most used tools [2]. This model is defined by the presence of at least three of the following criteria: weight loss, self-reported exhaustion, weakness, slowed gait speed, and lower energy expenditure. The pre-frailty condition is defined by the presence of one or two of these criteria, and a fit adult is defined as someone who does not meet any of these five criteria.

Frailty has been misleadingly associated with aging, comorbidity, or disability, although it does increase the risks of increased morbidity and lower survival. In addition, frailty is bidirectionally associated with chronic diseases. The presence of chronic conditions increases the risk of frailty [5], and the presence of frailty increases the risk of morbidity and mortality by accelerating the physical impairment of patients [6]. Therefore, a frailty assessment is of great importance in older adults, regardless of the main diagnosis [2].

Frailty, as a geriatric syndrome, is traditionally underdiagnosed outside the field of geriatric medicine. However, an increasing interest in inquiring about the relationship between chronic obstructive pulmonary disease (COPD) and frailty has been recently observed. COPD is one of the most prevalent and disabling chronic diseases in older adults, and it leads to a relevant decrease in survival [7]. The prevalence of frailty in older adults with COPD is high, even in younger populations (under 65 years old), ranging between 4% and 59% [8], although the use of the FFP decreases the prevalence down to 9.9% [8].

A significant increase in scientific works about the relationship between frailty and COPD has been observed in the last few years, in which the important prognostic role of a frailty diagnosis in the progression of COPD has been highlighted [9,10].

Likewise, COPD is the third leading cause of death worldwide. Many factors contribute to the development of COPD, including genetic factors (alpha1-antitrypsin deficiency), pollution, cigarette smoking, and occupational exposure to various chemicals. COPD manifests as an inflammatory disease that affects the airways, lung parenchyma, and pulmonary vasculature. Inhalation exposure can trigger an inflammatory response, leading to a decrease in forced expiratory volume and tissue destruction and eventually to airflow limitation. Airflow limitation is the main pathophysiological feature of COPD [11]. This relationship between the two may be because COPD and frailty share some risk factors, including aging, smoking, and inflammation [12] as well as clinical manifestations, such as fatigue, anorexia, muscle weakness, and slowed walking speed [13]. In this sense, Lahousse et al. suggested that both pathologies might have a single physiopathology in common [14].

Finally, it should not be forgotten that several common risk factors shared by COPD and frailty have been identified, including alcohol consumption, cigarette smoking, lower physical activity, poor diet, and older age [15]. All are related to functional and cognitive declines. For these reasons, an early diagnosis of frailty is needed to provide a specific and individualized care plan for the recovery and mitigation of the deleterious effects of frailty in these patients. Clinicians should actively screen frailty in COPD patients to improve their outcomes [15,16].

A systematic review and meta-analysis recently showed that the risk of frailty among patients with COPD diagnoses was twice as high as those of adults of the same age without COPD [17]. Both COPD and frailty share some risk factors, such as less favorable aging trajectories, nicotinism, and inflammation [18], along with clinical signs such as anorexia, fatigue, muscular weakness and slower gait speed (the latter three are the phenotypic criteria of frailty syndrome) [18], which could be due to their shared physiopathology [13].

However, the choice of the frailty assessment is one of the most common problems in frailty studies in different clinical settings. Given the increase in scientific literature on this topic, the authors of this paper considered it important to perform a narrative review of the literature in order to understand the actual prevalence of frailty in patients with COPD, the association of frailty with other geriatric syndromes in these patients, the influence of frailty on patient clinical evolution and survival, and the therapeutic approaches to reverse or reduce frailty in COPD patients.

## 2. Materials and Methods

The present review was carried out by conducting an electronic search in OVID^®^ (Ovid Technologies, Inc., Wolters Kluwer Health, NY, USA) (Medline and Embase) and PubMed^®^ (National Library of Medicine, Bethesda, MD, USA), combining the following MeSH keywords: “pulmonary disease, chronic obstructive”, and “frailty”. The search was completed on 30 September 2022 and limited to publications posted in the last 5 years; in English and Spanish; and in human subjects aged 60 or older. A total of 575 articles were obtained, of which 25 were finally selected. Details of the evaluation and selection process of the items are shown in Figure 1.

The articles were selected by four researchers based on the following eligibility criteria: The articles included in the criteria were meta-analyses, randomized clinical trials, cohort studies, case-control studies, observational studies, and before-and-after studies; the population needed for the review were patients 65 years old or older with an established COPD diagnosis; and prevalence and incidence studies using reported frailty assessment tools and reporting frailty prevalence were included, along with interventions to reverse frailty. It was stated in the exclusion criteria that letters-to-the-editor; case reports, narrative and systematic reviews without meta-analyses; articles with no available abstracts or those with only the abstract published; and studies that met the inclusion criteria but less than 50% of their study sample was under the age of 65 (i.e., predominantly non-geriatric), which indicated these were not in compliance with the aim of this review.

The review authors re-evaluated all articles, and the final inclusion was restricted to those of high enough quality with information relevant to the objectives of this review. The outcome measures examined were mortality, length of hospital stay, functional status, medical complications, destination after discharge, functional recovery, frailty reversion, readmissions, and survival. The authors used an Excel^®^ spreadsheet to record all the bibliographic references in order to identify and exclude duplicate papers. The Excel spreadsheet was also used to work with the selected articles and proceed to the selection of the articles. For this process, a bibliographic manager was used.

## 3. Results

The 25 selected articles for this review were classified according to their type: 1 meta-analysis, 1 cohort study, 1 retrospective cohort study, 8 longitudinal studies, 8 cross-sectional studies, 2 case-control studies and 4 retrospective studies. Most studies (22) reported frailty prevalence, which ranged from 6% to 85.9% and was partly related to the variability of the recruited patients’ age, ranging from 61.2 to 75.9 years, and most of the sample being male. There was a considerable variation in the frailty measurement tools, as up to 13 tools used in the studies included in the review were found: the hospital frailty risk score (HFRS) in two studies; the Fried frailty phenotype (FFP) in eight studies; the frailty index (FI) in four studies; the Kihon checklist (KCL) in two studies; the fatigue, resistance, ambulation, illnesses, and loss of weight (FRAIL) in two studies; the reported Edmonton frail scale (REFS), the Chinese–Canadian study of health and aging clinical frailty scale, the evaluative frailty index for physical activity (EFIP), the clinical frailty scale (CFS), the short physical performance test (SPPB), and the frail non-disabled questionnaire (FiND), in one study each; and the timed up and go (TUG) and the FI-lab in one study.

### 3.1. Frailty Prevalence

Out of the 25 selected studies, 22 of them included frailty prevalence data in the studied sample, which, as discussed in the previous section, ranged from 6% to 85.9%. The frailty prevalence reported by each study included in this narrative review are reported in Table 1.

Two studies, including a retrospective cohort study [20] and a prospective propensity score-match study [21] that used the HFRS as a frailty screening tool, showed a disparate prevalence of 14% and 57%, respectively, in samples of similar age. The former was conducted in hospitalized patients, with 68.7% of the sample having the highest degree of dyspnea (score 4 and 5) estimated with the Hugh–Jones dyspnea scale score while the latter had no data on disease severity.

However, the most widely used tool for the diagnosis of frailty was Linda Fried’s phenotypic criteria, as it was used in eight COPD studies [22,23,24,25,26,27,28,29]. Thus, when using the FFP, the prevalence of frailty was 6% [23] and 67% [24], respectively. The ages of the participants in these two studies were similar. In the first study, which used FFP to detect frailty, patients had to have moderate or severe COPD while the second study was performed in patients admitted for an acute exacerbation of COPD. Of the 8 studies that used FFP as a screening tool for frailty, the second with the highest prevalence (49.8%) was completed in patients almost 20 years older than the 2 previous studies and who were diagnosed with COPD based on the 2017 Global Initiative for COPD (GOLD) guidelines. They had respiratory symptoms or risk factors and a post-bronchodilator ratio of forced expiratory volume in 1 s (FEV1) to forced vital capacity (FVC) < 0.70. The study by Kennedy et al. [22] reported an estimated incidence of frailty of 6.4 per 100 human years and the presence of frailty being associated with a lower quality of life. The fourth study that used FFP for the diagnosis of frailty had patients of an age similar to the first described with these criteria and showed a prevalence of frailty of 23% in patients with a diagnosis of COPD (post-bronchodilator FEV1/FVC < 80% and FEV1% predicted < 80%) and who also had been stable for at least the last 4 weeks [25]. A prospective study conducted in 119 patients with stable COPD during a 2-year follow-up assessed transitions between non-frail, pre-frail, and frail conditions using the FFP. A frailty prevalence of 7.6% and a pre-frailty prevalence of 73.1% were reported at the beginning of the study. After 2 years, 11.7% of patients worsened, 17.6% improved, and 70.5% remained the same. The variables associated with deterioration were dyspnea, disability, and lower handgrip strength, while those who improved presented greater handgrip and quadricep strength. The authors concluded that frailty is a dynamic condition, and therefore, the transitions between the states of frailty are related to significant changes in the clinical outcomes [26]. Furthermore, the authors of this narrative review designed a cross-sectional study in a prospective cohort of 127 patients with stable COPD (diagnosis based on the GOLD 2017 guideline) and classified 24.4% of the recruited sample as frail. The variables with independent associations with frailty included the mMRC, HAD-DEP, and age [27]. The last two articles that used FFP as a frailty screening tool compared this measure to the IF [28] and to the IF, SPPB, and CFS [29], respectively. The first analyzed data extracted from the UK Biobank of patients with COPD, who had been identified by linked primary care data, using a previously validated list of diagnostic codes and a severity assessment of the disease that was measured with spirometry data (20% of patients with FEV_1_ < 50%) and then compared two frailty measurements, the FFP and the FI, during an 8-year follow-up period. Frailty prevalence was high regardless of the frailty assessment tool used (17% frail according to the FFP and 28% moderate and 4% severely frail according to the FI), but only the FFP was associated with a lower FEV_1_ [28] while the second, a prospective study with a 2.18-year [IQR 1.56–2.62] follow-up of 302 COPD patients (diagnosis based on the GOLD 2017 guidelines), assessed the predictive capacity of 4 frailty scales: FFP, CFS, FI, and SPPB. Frailty prevalence was lower when using the FI (18%) and higher when using the other three tools (FFP, 51%; SPPB, 58.6%; CFS, 64.2%) [29].

Two papers measured frailty prevalence with FRAIL [30,31]. The first one, a case-control study [30], found a frailty prevalence of 25.3% in COPD patients (FEV1 to forced vital capacity of <0.7) assessed by the outpatient department and measured with the FRAIL scale, and these were linked to a higher prevalence of geriatric syndromes in COPD patients. The second one, an observational cross-sectional study in stable COPD patients (FEV1/FVC ratio of <0.7), found a greater risk of frailty in the GOLD groups B and D. Frailty was significantly correlated to the COPD assessment test score (CAT) and the MRC. The authors reported that the combination of CAT/MRC ≥ 5.5 was associated with the presence of frailty (odds ratio (OR) 6.7; 95% CI: 3.2–13.9) [31]. Two articles from the same research group used the IF as a diagnostic tool for frailty [32,33]. The first one, a cross-sectional study conducted in 520 community-based patients with COPD (confirmed by spirometry at entry), showed that patients with COPD presented a higher likelihood in the FI (0.16 (SD 0.08)) than the controls (0.05 (SD 0.03)), and the frailty prediction factors were the distance covered in a 6-minute walk distance (6MWD), the number of comorbidities, handgrip strength, and the number of acute exacerbations [32]. The second one, a case-control study, assessed the possible usefulness of the timed-up-and-go (TUG) tool for frailty detection. COPD patients presented a higher FI and a higher time using TUG (11.55 (SD 4.03 s)) than the controls (9.2 (SD 1.6 s)) [33]. This difference remained after adjusting for age and pulmonary function. The frailty predictive capacity using TUG (a cut-off point of 0.09 for the FI and 8 s for the TUG) was OR 2.67 (95% CI: 1.5–4.6) [33].

A retrospective study assessed the possible usefulness of a frailty evaluation tool (FI-lab) in COPD patients (primary clinical diagnosis of AECOPD at hospital admission) distributed between survivors and deceased. The FI-lab values were classified into 4 groups: <0.2, 0.2–0.4, 0.4–0.6, and >0.6. FI-lab < 0.4 values were seen in 88.3% of survivors while 75.3% of the deceased presented those values. The difference in the FI-lab was statistically significant between survivors (0.5 (SD 0.1)) and deceased (0.3 (SD 0.1)), *p* < 0.001 [34]. Two papers used KLC as a frailty detection tool in COPD patients (GOLD 2021) [35,36]. The first one, a cross-sectional study, found a frailty prevalence of 50% and reported negative effects of COPD on the central nervous system, along with depressive symptoms [36]. At the same time, a cross-sectional study conducted in 128 COPD patients found a frailty prevalence of 37.5% and estimated the relationship between the patient-reported outcome measures for dyspnea-related behavior, activity limitation (PROMs-D, which was consistent between the activity-limiting dyspnea scale (ADS) and the self-limiting dyspnea scale (SDS)), and frailty. Both ADS and SDS presented a high predictive capacity for frailty, although it was the PROMs-D (the sum of ADS and SDS) that was the most effective measure to classify frailty. For this reason, the authors found that the PROMs-D could be used as a frailty screening measure in patients with COPD [36].

A cross-sectional study conducted in COPD patients that lived in the community (posing the question, “Has a medical doctor ever told you that you have any of the following conditions: emphysema, chronic bronchitis, or chronic obstructive lung disease?”) analyzed the prevalence of geriatric syndromes in the recruited sample, including frailty (at 16%, evaluated according to the modified frailty scale) [37]. A prospective observational study conducted in hospitalized patients with acute exacerbations of COPD (AECOPD) reported a prevalence of moderate or severe frailty of 35.9% (18.4% severe frailty) using the REFS tool within 48–96 h of hospital admission [38]. At the same time, a cross-sectional study recruited 125 patients with COPD and assessed frailty using the Chinese–Canadian Study of health and aging clinical frailty scale and dyspnea using the modified medical research council questionnaire. The patients were divided into two groups: dyspnea and non-dyspnea. The prevalence of frailty in the non-dyspnea group was 26.7%, and it was 85.9% in the dyspnea group. The predicting factor of frailty in the dyspnea group was the number of prescribed drugs, and both the polypharmacy and the CAT were positively correlated with the conversion time from fit to frail in both groups [39].

A cross-sectional study of 57 patients with COPD (GOLD 1–4) had the highest prevalence of frailty (EFIP) detected in this narrative review at 83%, which was associated with malnutrition and physical frailty [40]. On the other hand, the prevalence of slight-to-moderate frailty assessed using the CFS was 54% in 46 patients admitted to the hospital due to AECOPD [41]. Finally, similar results were reported in a study in primary care. Thus, a cross-sectional study conducted in 257 COPD patients in primary care assessed the frailty determinants of these patients. A frailty prevalence of 82% was reported using the FiND questionnaire. The risk of presenting frailty was associated with age, hypertension, uncontrolled disease (CAT ≥ 10), and mMRC ≥ 2, or the presence of ≥2 AECOPD and GOLD status (B and D vs. A and C groups). The authors concluded that the severity of COPD increased frailty prevalence [42].

This frailty prevalence was consistent with the one published in a meta-analysis (27 studies: 23 were cross-sectional, 3 were longitudinal, and 1 was mixed (both cross-sectional and longitudinal)). These established a collective prevalence of frailty of 20% in COPD patients (95% CI: 15–24%; I^2^ = 94.4%) while pre-frailty prevalence was 56% (95% CI: 52–60%; I^2^ = 80.8%). Patients with COPD showed an OR of presenting frailty of 1.97 (95% CI: 1.53–2.53) [17].

#### The Prevalence of Geriatric Syndromes Linked to Frailty in COPD Patients

The frailty prevalence in COPD patients was linked to a high prevalence of geriatric syndromes, as four of the studies included in this narrative review showed. Thus, malnutrition and frailty coexisted in 40% of the COPD patients, malnutrition and physical frailty coexisted in 18%, and malnutrition and disability coexisted in 21% of the cases in a cross-sectional study [40]. This overlap of geriatric patients indicates the need for nutritional intervention in COPD patients, especially before starting a rehabilitation program [40]. The EFIP and PG-SGA scores were significantly correlated (r = 0.43, *p* = 0.001), as well as the Fried’s criteria and the PG-SGA score (r = 0.37, *p* = 0.005). A nutritional intervention should be delivered by health care professionals in COPD patients before starting a rehabilitation program [40]. A case-control study [30] assessed the prevalence of geriatric syndromes in 150 COPD patients. Frailty was linked to a high prevalence of geriatric syndromes, such as impairment in instrumental activities of daily living (IADL), 37.3%; cognitive impairment, 35.3%; urinary incontinence, 20.7%; and malnutrition (20.7%). Two variables increased the risk of the prevalence of geriatric syndromes: dyspnea (≥2 mMRC grade) and low socioeconomic status [30]. The authors emphasized the importance of frail COPD patient performance in the comprehensive geriatric assessment (CGA). Frailty has been associated with mood disorders in COPD patients. A cross-sectional study of 40 COPD patients reported a low perception of quality of life. In addition, patients classified as frail using the KCL score presented lower left and right hippocampal, subiculum, and presubiculum volume, as compared to non-frail patients. The authors concluded by pointing to the impact of frailty on the hippocampal volume and its combined associations with a poor quality of life [35]. Finally, a cross-sectional study conducted in 3222 older adults (69.6 (SD 7.4) years) with COPD who lived in the community analyzed the prevalence of geriatric syndromes in the recruited sample, including frailty, functional disability, physical function impairment, lower physical activity, falls, polypharmacy, loneliness, depression, cognitive impairment, urinary incontinence, and comorbidity [37]. The prevalence of geriatric syndromes was higher in COPD patients than in non-COPD patients: 16%, frailty; 53.9%, urinary incontinence; 57.7%, loneliness perception; 58.1%, functional disability; 12.9%, moderate cognitive impairment; 32%, depressive symptoms; and 37.5%, severe polypharmacy (>10 prescribed drugs). As compared to non-COPD patients, the frailty-adjusted OR was 6.3 (95% CI: 3.0, 13.0). This meant that the COPD patients had a risk of frailty more than six times higher than the controls. Likewise, the adjusted OR for functional disability was 1.4 (95% CI: 1.01, 2.0); for impaired physical function, the adjusted OR was 2.1 (95% CI: 1.1, 3.7); for extreme low physical activity, the adjusted OR was 2.3 (95% CI: 1.5, 3.5); for polypharmacy (≥10 medications), the adjusted OR was 2.9 (95% CI: 2.0, 4.2); and for depression, the adjusted OR was 1.9 (95% CI: 1.4, 2.7) [37].

### 3.2. Frailty as a Predictive Factor of Poor Outcomes

Out of the 25 articles selected for this review, 12 associated the presence of frailty with worsening outcomes, such as AECOPD, hospitalization, length of stay, greater difficulty in returning home, hospital readmissions, and mortality. Table 2 summarizes the association between frailty and poor outcomes in COPD patients, as reported by the studies included in this narrative review.

### 3.3. Frailty as a Predictive Factor of Poor Outcomes

Out of the 25 articles selected for this review, 12 of them associated the presence of frailty with worsening outcomes such as AECOPD, hospitalization, length of stay, greater difficulty in returning home, hospital readmissions, and mortality. Thus, in a retrospective cohort study [20], frailty was associated with an increase in hospitalization (32.9% vs. 17.5%), in-hospital mortality (16.4% vs. 12.5%), greater difficulty in returning home (34.6% vs. 22.9%), and a poorer quality of life at discharge (8.7% vs. 12.4%) [20]. The higher HFRS score was independently associated with a prolonged hospital admission (length of stay ≥ 30 days) (OR 2.0; 95% CI: 1.4–2.9) [20]. The ability of the HFRS to predict prolonged hospitalization was slightly higher than that of the Charlson comorbidity index (CCI) [21]. In a similar way, the severely frail patients were also much more likely to be readmitted than the non-frail patients (45% vs. 18%), and after adjusting for age and relevant disease-related factors, severe frailty remained an independent risk factor for 90-day readmission (OR = 5.19; 95% CI: 1.26–21.50) [38]. Frail participants also had an increased rate of hospitalization (adjusted hazard ratio (HR), 1.6; 95% CI: 1.1–2.5; *p* = 0.02), an adjusted increase in hospital use of 8.0 days (95% CI: 4.4–11.6; *p* = 0.001), and a higher mortality rate (adjusted HR 1.4; 95% CI: 0.97–2.0; *p* = 0.07) [22]. Likewise, after adjustment, frailty increased the incidence of AECOPD (IRR = 1.75, 95% CI: 1.09–2.82) and all-cause hospital admissions (IRR = 1.39, 95% CI: 1.04–1.87). All-cause mortality risk also increased during the 1-year follow-up, which was higher in frail patients (HR = 2.54, 95% CI: 1.01–6.36) [24].

However, in another study, the global frailty rate was not associated with the incidence of AECOPD, although the estimated weakness component was associated with a higher risk of AECOPD, which was assessed by handgrip strength (IRR 1.46, 95% CI: 1.09–1.97); moreover, frailty was associated with the risk of having non-COPD hospital admissions [25]. In contrast, the FI-lab scores statistically increased the risk of AECOPD and mortality [41], and greater frailty severity was associated with an increase in care costs, a longer hospital stay, more previous hospitalizations, and subsequent hospitalizations with an “alternate level of care” (ALC) [41].

It was previously mentioned that weakness, established by handgrip strength performance, was a predictor of AECOPD. This indicator was also present in another study that had a 30-day readmission predictive capacity (OR 11.2; 95% CI: 1.3–93.2) [23]. Frailty assessed by the FFP was associated with a higher risk of mortality (HR 2.3; 95% CI: 1.8–3.0), major adverse cardiovascular events (MACE) (HR 2.7; 95% CI: 1.7–4.5), hospital admissions (HR 3.4; 95% CI: 2.8–4.1), hospital exacerbations (HR 5.2; 95% CI: 3.8–7.1), and community exacerbations (HR 2.1; 95% CI: 1.8–2.5). The FI (severe frailty compared to the robust condition) obtained similar outcomes as to mortality (HR 2.6; 95% CI: 1.7–4.0), MACE (HR 6.8; 95% CI: 2.7–17.0), hospital admissions (HR 3.7; 95% CI: 2.5–5.4), hospital exacerbations (HR 4.3; 95% CI: 2.4–7.7), and community exacerbations (HR 2.4; 95% CI: 1.7–3.3) [28]. Similarly, in a different study, all assessed scales (FFP, FI, CFS, and SPPB) were associated with an increase in 1-year mortality [29]. During the follow-up, all tools, except for the FI, were associated with mortality in a multivariate analysis: FFP, HR = 3.11 (95% CI: 1.30–7.44); CFS, HR = 3.68 (95% CI: 1.03–13.16); and SPPB, HR = 3.74 (95% CI: 1.39–10.06). The FFP was associated with AECOPD, and the FFP, the CFS, and the FI-CD were associated with the number of hospital admissions [29]. The CFS and the FI showed a sensitivity of 96% for predicting all-cause 1-year mortality in COPD patients, but all scales showed low specificity that ranged from 39% to 44% [30]. The receiver-operating characteristic (ROC) curves for the four frailty scales each had a similar capacity for predicting mortality, presenting no statistically significant differences. When associating variables, such as age, sex, medication, CCI: GOLD severity, and CAT, the tools improved their capacity to predict mortality [29].

The last two papers included in this section showed that frailty was associated with more COPD exacerbations, being 2.2 (SD 1.7) in frail patients and 1.0 (SD 1.0) in fit COPD patients [27]; at the same time, a longitudinal study conducted in 2706 patients (76.4 years) assessed the link between the presence of frailty, which was estimated with the FI, and the presence of COPD, and highlighted that the presence of both conditions increased 3-year mortality risk up to a 95% (HR 1.95; 95% CI: 1.18–3.2) [43].

### 3.4. Interventions in Frailty COPD Patients

Only one article focusing on frailty intervention was found in this narrative review. The intervention consisted of an integrative multidisciplinary approach that included evidence-based pharmacological and non-pharmacological interventions, such as palliative, respiratory, and rehabilitative therapies. The physical intervention consisted of physiotherapy and occupational therapy 4–5 times a week, each session lasting from 30 to 45 min, in which limb strengthening and aerobic activities were performed and tailored to the individual baseline function. In this study, 100 older-adult-matched pairs were recruited (73.9 (SD 8.2) years). High HFRS was present in 57%, and 71% had overlapping respiratory diagnoses. In this sample, integrated care for advanced respiratory disorders was associated with a further reduction in the length of hospital stay, down to 9.1 (SD 19.9) days; fewer admission days, 0.8 (SD 1.9); and fewer ED visits, 0.6 (SD 2.2). The 6MWD and the MBI scores improved. Greater improvement was observed in patients with lower baselines in their 6MWD and MBI scores. Prescriptions of slow-release opioids rose from 9% to 49%, and treatments for anxiety and depression rose from 5% to 19% [21].

## 4. Discussion

In the present review, the authors observed a significant variability in the frailty measurement tools used, in the associations among the presence of frailty, a high prevalence of geriatric syndromes and COPD-related aspects, in the close relationship between the presence of frailty and poorer health outcomes, and finally, in a single study that explored the potential benefits of a combined intervention (pharmacological and non-pharmacological, including rehabilitation and a nutritional approach), which showed improvements in functional parameters.

Despite having a generally accepted definition of frailty [1], the number of operational frailty definitions was high [44], along with the number of frailty assessment tools, which presented a significant heterogeneity for the validation in different settings [4]. This exact situation was observed in the review, in which up to 13 different measurement tools used for the assessment of frailty in COPD patients were identified. Only one study [29] compared frailty prevalence and its prognostic capacity for adverse health outcomes according to four scales (FFP, CFS, FI, and SPPB). The FI presented the lowest frailty prevalence, but the rest (FFP, CFS, and SPPB) presented similar outcomes. The FI is an index based on the accumulation of deficiencies and detects more advanced severity than the FFP. In fact, as the authors of a cross-sectional study highlighted, different frailty instruments may capture overlapping, albeit distinct, parameters, and thus, they should not be used interchangeably [45]. Therefore, the prevalence of frailty was lower when using the FFP (17%), as compared to the FI [28]. Although when using four tools (FFP, 51%; SPPB, 58.6%; FI-CD 59.6%; CFS, 64.2%), the authors reported a moderate-to-substantial agreement between the instruments [29]. Despite the concordance reported in a previous study [30], the heterogeneity in the assessed domains of the different frailty tools [4], and the severity of COPD, the prevalence of frailty was inconsistent among the studies included in this review, regardless of the stability of the COPD, the setting where the patients were recruited, the age range of the sample, or the instrument (even within the same instrument). Nevertheless, the prognostic capacity of frailty in patients with COPD persisted despite the variance in ages, instruments, and the degrees of severity of COPD, confirming the importance of frailty as an indicator of disease progression in these patients.

The only meta-analysis included in this review [17] highlighted that the risk of frailty in COPD patients was 97%, and it had important clinical implications, which required the use of the CGA in patients with positive frailty screenings to order to establish appropriate programs for reversing or reducing frailty. The authors of the aforementioned meta-analysis emphasized the common mechanisms and risk factors of COPD and frailty as well as the increased risks of mortality and hospital readmissions in the acute exacerbation of COPD in frail patients. However, the relationship among frailty, geriatric syndromes, and COPD were not assessed in the meta-analysis, as reflected in this narrative review, nor were other types of adverse events associated with frail COPD patients reported, as described in this narrative review.

Likewise, frailty has been related to the severity of COPD [20]. Since frailty prevalence is high in COPD patients, regular assessments of frailty in clinical practices with COPD patients should be performed to prevent or mitigate the progression of the disease [42]. Since the CGA is time-consuming, the TUG could be used as a screening tool [33] due to its ability to predict frailty in COPD patients.

Frailty prevalence increases with each decade of life. In fact, the only meta-analysis included in the study stated that age was one of the variables that increased the risk of frailty. However, most of the prevalence risks reported in these studies were higher, as compared to those measured in populations of similar age without COPD. Some published studies have linked the higher prevalence of frailty to the severity level of COPD and found a direct association using such measurements as mMRC [27,30,31,42], CAT [31,39,42], GOLD [31,42], FEV_1_ [28], PROMs-D (ADS plus SDS) [36].

Just as the progression of diseases such as HIV/AIDS and higher rates of inflammatory activity increased frailty prevalence, as compared to patients of the same age without these diseases [46], the prevalence of frailty in COPD patients was also higher, which was likely due to higher inflammatory activity. Therefore, it would be interesting to explore whether pathophysiological factors (Figure 2) or possible adverse effects of drugs given for active disease management, such as corticosteroids, may accelerate the onset of frailty and other geriatric syndromes such as sarcopenia, which is an important element in the functional performance in older adults. As a recent review described, beyond chronic inflammation and reduced physical activity, factors that decrease muscle strength and endurance in COPD patients include oxidative stress, inactivity, hypoxemia, hormone abnormality, lack of nutrients such as protein and vitamin D, and the use of corticosteroids [15].

Despite the high prevalence of sarcopenia in COPD patients (15.5%) and its relation to the severity of pulmonary disease, the FEV_1_, poor exercise tolerance, and poor quality of life [47], none of the studies included in the review assessed the presence of sarcopenia in their samples.

The presence of frailty has been related to the high prevalence of geriatric syndromes in COPD patients. Malnutrition, cognitive impairment, polypharmacy, disability, and depressive symptomatology have been frequently reported in these patients. Consequently, various studies have highlighted the importance of early detection and using the CGA for the global assessment of these patients, with the aim of defining individualized care programs for the improvement of their clinical and functional outcomes. Thus, six of the included articles linked the presence of frailty in COPD patients with a high prevalence of other geriatric syndromes, such as malnutrition [30,40], IADL impairment [30], cognitive impairment [22,30,35,36], urinary incontinence [28,30], physical disabilities [22,26,28], depression [27,28,35], polypharmacy [28], falls [28], loneliness [28], and poor quality of life [22,35]. Given the increased presence of geriatric syndromes in older adult patients with COPD, the authors of various studies have highlighted the importance of the regular use of the CGA [22,32] for a complete multidimensional assessment that detects potentially correctable impairments and comorbidities, with the aim of reducing mortality rates [32]. An approach “beyond the lung” [37] is necessary in the care of these patients and should be focused on the management of geriatric syndromes and conditions, as their improvement enhances patient quality of life and clinical COPD outcomes.

Frailty and comorbidity are predictors of a worsening progression in patients under different clinical conditions [48]. A study in the present review stated that frailty was a better predictor of adverse events than the CCI [20]. Frailty, as a recent review highlighted, presents limited evidence regarding the increase in morbidity and mortality in COPD patients [49], as it is a better indicator of functional capacity and the need for palliative care in the future than a prognostic indicator. However, frailty has been described as a factor linked to the higher incidence of both adverse events [50] and higher mortality [51] in adults that live in a community. Similarly, several studies included in this review emphasize frailty as a risk factor of worsening outcomes, such as AECOPD [20,25,27,28,34,39], hospitalization [20,24,25,28,29,41], length of stay [20,24,39], greater difficulty in returning home [20], hospital readmission [20,23,24], MACE [28], medical costs [41], transition to a higher level of care [41], and mortality [20,24,28,29,34,43]. The authors of one of these studies [43] emphasized that vulnerability caused by frailty increased adverse outcomes in older adults, and the increment of a worsening prognosis for frailty in these patients requires regular assessment in clinical practice. As previously mentioned, frailty was a predictor of worsening health outcomes [50] and mortality [51] in older adults, as well as in older adults with COPD [20,22,23,24,25,27,28,29,34,38,41,43]. In fact, COPD patients who were also frail had worsening outcomes than COPD patients without frailty. Therefore, we can consider frailty an indicator of a worsening progression of COPD, and it should be detected early using a CGA in order to slow or reverse its impact.

The CGA is completed with the design of an individualized care plan, in which non-pharmacological measures are suggested to reverse or reduce present geriatric syndromes. Only one study focused on frailty intervention was found in this narrative review [21]. The physical intervention improved the functional performance of the recruited patients, leading the authors to highlight that the integration of functional rehabilitation with palliative care could improve the functional capacity of patients, along with better treatments for symptoms such as anxiety and depression, which are typically reported. These findings are consistent with some suggestions outlined in a recent review [52] that highlighted that the ability to reverse frailty without intervention was minimal but still possible through pulmonary rehabilitation (PR) programs, which also improved the prognosis of these patients. However, the mechanism by which PR reversed frailty in this patient population was not elucidated. PR programs with multidisciplinary components could reverse frailty by addressing the five components of the frailty phenotype; however, the heterogeneity of the COPD population hampers the uniformity of scheduled exercise programs, as well as the objectives. Once again, it would be necessary to individually design programs according to the patient’s outcomes based on their CGA. Likewise, another review [53] highlighted the significant association between frailty and COPD that requires the early detection and treatment of frailty in order to reduce the risk of worsening health outcomes, such as increased functional impairment, disability, hospital admissions, presence of geriatric syndromes, institutionalization, and death. Similarly, a review that included 20 scientific papers that described interventions using pulmonary rehabilitation, electrical stimulation, home-based programs, geriatric rehabilitation, hospital-based exercises, physical activity, and non-standardized exercise, reported that different programs were successful when they sought a therapeutic partnership with the patient, were individualized, increased patient engagement, and improved robustness and adaptive capacity [54]. Building trust, individualizing priorities, and approaching multidimensional problems appropriately was necessary for the success of the programs. Home-based pulmonary rehabilitation was another method by which to treat frailty in these patients in order to improve functional capacity, frailty status, quality of life, and pulmonary symptoms, such as fatigue [55]. Palliative care should not be overlooked. As a recent review noted, palliative care is much more than hospice or end-of-life care [56] and could be integrated with the above-mentioned interventions and approaches to provide a holistic, comprehensive assessment and treatment, pharmacological and non-pharmacological, to manage symptoms and enhance quality of care for these patients.

Lastly, another review highlighted the need to improve the comprehension of the frailty phenotype in different chronic diseases, given its high prevalence in COPD patients. From a clinical perspective, it requires interdisciplinary cooperation in order to mitigate the impact of frailty in COPD patients [57]. In this sense, and given the particular characteristics of frail patients with COPD, not only would it be necessary to establish a “COPD–frail” phenotype but also to develop specific interventions in order to reduce the prevalence of geriatric syndromes and improve functionality, quality of life, and survival while reducing acute exacerbations and hospitalizations. Certainly, the authors were unable to determine through this narrative review which frailty tool was optimal for patients with COPD or whether it was necessary to use several scales simultaneously (nor could the authors suggest the appropriate combination of instruments) to improve diagnostic sensitivity and specificity. Once again, given the importance of frailty in the clinical course of COPD patients, the authors considered the need to emphasize the standardization of frailty screenings in both outpatient and inpatient settings. It could increase the inclusion of patients when selecting the appropriate interventions. It could improve or reverse frailty and other associated geriatric syndromes as well as the evolution of the disease (reducing AECOPD, hospital admissions, and mortality), and it could maintain or improve the quality of life in COPD patients. Though the discussion section is appropriate for any speculation concerning new hypotheses, such hypotheses need to be ratified by relevant studies that, at best, confirm them or, at worst, refute them.

The limitations of this review stem from the absence of randomized clinical trials and a considerable number of cross-sectional studies that identified associations but not causalities. The authors evaluated the published articles in the OVID and PubMed databases in order to include articles published in journals with the greatest impact. However, this decision was not unbiased, as publications in other databases that could have been of interest were not evaluated. Likewise, the authors are aware that a systematic review could have provided more evidence on the topics covered. However, the heterogeneity of the frailty assessment instruments, the different settings in which the patients were recruited, and the varied degrees of severity of disease in the selected studies led the authors to dismiss this possibility. Given the growing interest in this topic, the authors of this review are considering the development of a systematic review in a near future.

## 5. Conclusions

Frailty prevalence is high in COPD patients. The heterogeneity of frailty measurement tools challenges drawing broad conclusions about the results obtained with such tools. Regardless of the assessment tool used, frailty prevalence in COPD patients has been associated with a high prevalence of geriatric syndromes and worsening clinical outcomes, including mortality. It is recommended to use frailty screenings for COPD patients, regardless of the setting in which they are assessed, and to perform CGAs in order to detect associated problems and to establish individualized treatment plans in order to improve clinical outcomes in these patients.

## Figures and Tables

**Figure 1 ijerph-20-01678-f001:**
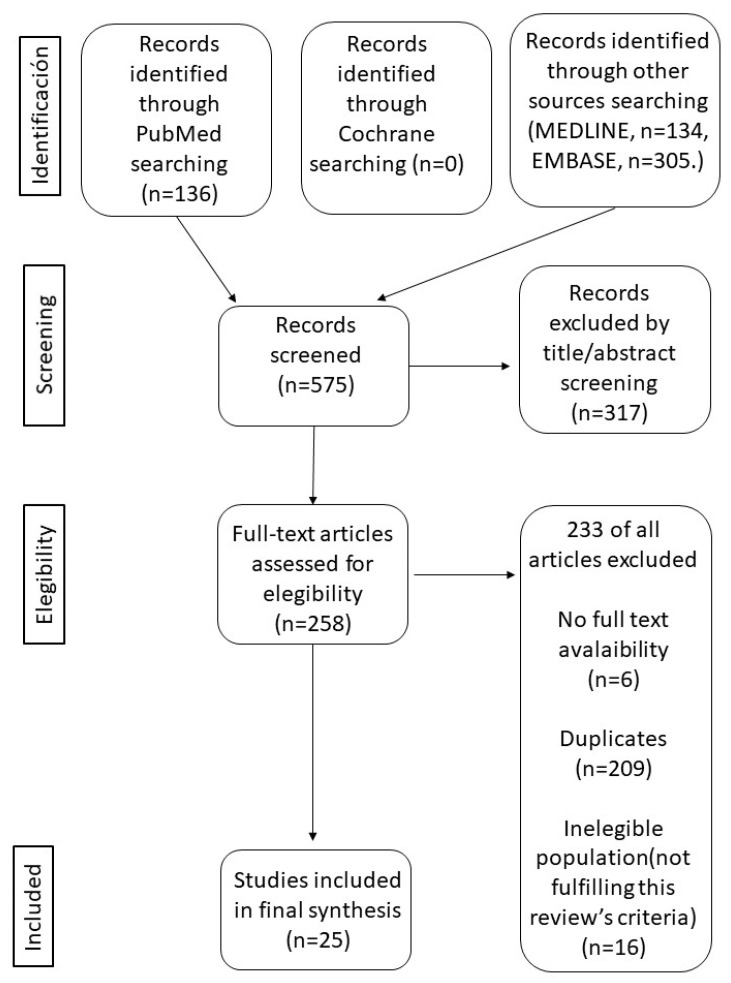
PRISMA flow diagram for database search and study selection process. Based on Moher et al. [19].

**Figure 2 ijerph-20-01678-f002:**
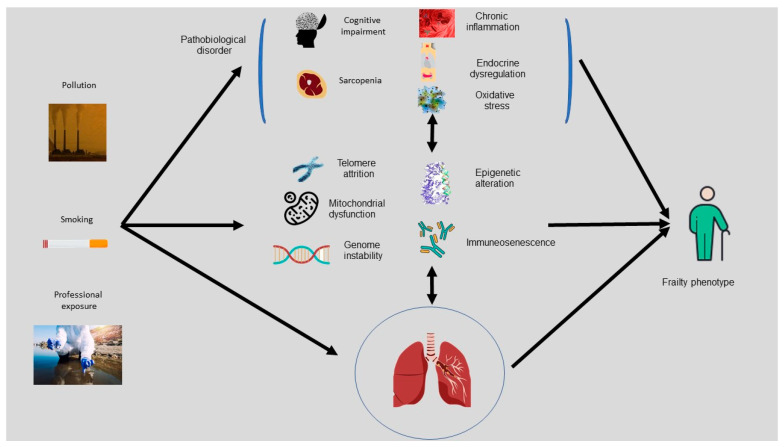
Graphic description of the link between environmental factors, pathobiological disorders, genetic factors, immunosenescence, and frailty.

**Table 1 ijerph-20-01678-t001:** Summary of studies with reported prevalence of frailty in COPD patients.

Article	Type	Sample	Age (Years)	Tool Used	Frailty Prevalence	Spirometric COPD Confirmation
Ushida, K. et al. [20]	Retrospective cohort	3396 COPD patients	75.9 (SD 11.2)	HFRS	14%	No (ICD-10 codes: J41–J44)
Neo, H.Y. et al. [21]	Prospective, propensity score-match study	100 matched pairs	73.9 (SD 8.2)	HFRS	57%	No
Kennedy, C. C. et al. [22]	Retrospective cohort	902 COPD patients	67 [IQR 63–70]	FFP	6%	Yes
Witt, L. J. et al. [23]	Observational study	70 patients admitted due to COPD exacerbation	63.5 (SD 58.1, 71.3)	FFP	67%	Yes
Luo, J. et al. [24]	Cross-sectional study	309 COPD patients	86 [IQR 80–90]	FFP	49.8%	Yes
Yee, N. et al. [25]	Prospective cohort	280 COPD patients	68.6 (SD 9.2)	FFP	23%	Yes
Bernabeu-Mora, R. et al. [26]	Prospective study	119 COPD patients	66.9 (SD 7.9)	FFP	7.6%	Yes
Naval, E. et al. [27]	Cross-sectional study	127 COPD patients	66.5 (SD 7.9)	FFP	24.4%	Yes
Hanlon, P. et al. [28]	Observational study	3132 COPD patients	61.9 (SD 5.9)	FFP FI	17% FFP 32% FI	Yes
Zhang, D. et al. [29]	Prospective study	302 COPD patients	86 [IQR 80–90]	FFP CFS FI-CD SPPB	FFP 51% CFS 64% FI 58.6% SPPB 59.6%	Yes
Soni, N. et al. [30]	Case-control study	150 COPD 150 Controls	65.98 (SD 5.43) 65.72 (SD 5.65)	FRAIL	25.3%	Yes
Dias, L. S. et al. [31]	Cross-sectional study	150 COPD patients	67.0 (SD 61.0–71.5)	FRAIL	50.3%	Yes
Gale, N. S. et al. [32]	Case-control study	520 COPD 150 controls	(66.1 (SD 7.6)) (65 (SD 7.4))	FI	28%	Yes
Albarrati, A. M. et al. [33]	Case-control study	520 COPD 120 controls	66.1 (SD 7.6) (65 (SD 7.4))	FI	76%	Yes
Gu, J. J. et al. [34]	Observational retrospective study	154 COPD patients	79.73 (SD 8.38)	FI-lab	75.3%	Yes
Takahashi, S. et al. [35]	Cross-sectional study	40 COPD patients	70.6 (SD 8.21)	KCL	50%	Yes
Oishi, K. et al. [36]	Observational study	128 COPD patients	73 [IQR 69–78]	KCL	37.5%	Yes
Witt, L. J. et al. [37]	Cross-sectional study	322 COPD patients	69.6 (SD 7.4)	modified frailty	16%	No
Bernabeu-Mora, R. et al. [38]	Prospective cohort	103 hospitalized COPD patients	71 (SD 9.1)	REFS	35.9% moderate or severe frailty	Yes
Chen, P. J. et al. [39]	Cross-sectional study	125 COPD patients	77.36 (SD 10.26)	Chinese Canadian study of health and aging clinical frailty scale	85.9% dyspnea group 26.7% non-dyspnea group	Yes
Ter Beek, L. et al. [40]	Cross-sectional study	57 COPD patients	61.2 (SD 8.7)	EFIP	83%	Yes
Chin, M. et al. [41]	Prospective study	46 patients admitted due to COPD exacerbation Mild frailty Moderate frailty Severe frailty	72 (SD 9) 72 (SD 10) 76 (SD 12)	CFS	54%	No described
Ierodiakonou, D. et al. [42]	Cross-sectional study	257 COPD patients	(65 (SD 12.3))	FiND (frail non-disabled)	82%	Yes

Legend: COPD = chronic obstructive pulmonary disease; HFRS = hospital frailty risk score; EFIP = evaluative frailty index for physical activity; FFP = Fried frailty phenotype; FI = frailty index; REFS = reported Edmonton frail scale; FRAIL = the fatigue, resistance, ambulation, illnesses, and loss of weight; KCL = Kihon checklist; Chinese–Canadian Study of health and aging clinical frailty scale; FI-lab = frailty index based on deficits in laboratory test; CFS = clinical frailty scale; SPPB = short physical performance test; FiND = the frail non-disabled questionnaire; SD = standard deviation; IQR = interquartile range.

**Table 2 ijerph-20-01678-t002:** Relationship between health outcomes and frailty in COPD patients.

Article	Type	Sample	Poor Outcomes Associated with Frailty
Ushida, K. et al. [20]	Retrospective cohort	3396 COPD patients	Hospital admissions (32.9% vs. 17.5%) In-hospital mortality (16.4% vs. 12.5%) Greater difficulty in returning home (34.6% vs. 22.9%)
Kennedy, C. C. et al. [22]	Retrospective cohort	902	Increased rate of hospitalization: Adjusted HR, 1.6 (95% CI: 1.1–2.5) Increase in hospital use of 8.0 days: (95% CI: 4.4–11.6) Higher mortality rate: Adjusted HR 1.4 (95% CI: 0.97–2.0); *p* = 0.07
Witt, L. J. et al. [23]	Observational study	70 patients admitted due to COPD exacerbation	30-day readmissions: OR 11.2 (95% CI: 1.3–93.2)
Luo, J. et al. [24]	Cross-sectional study	309	AECOPD: IRR = 1.75 (95% CI: 1.09–2.82) All-cause hospitalizations: IRR = 1.4 (95% CI: 1.0–1.9) All-cause mortality risk: HR = 2.5 (95% CI: 1.0–6.4)
Yee, N. et al. [25]	Cohort study	280	Handgrip strength increased AECOPD risk: IRR 1.46 (95% CI: 1.09–1.97)
Naval, E. et al. [26]	Cross-sectional study	127	AECOPD: Frail COPD patients 2.2 (SD 1.7) vs. fit COPD patients 1.0 (SD 1.0)
Hanlon, P. et al. [27]	Observational study	3132	FFP Mortality risk: HR 2.3 (95% CI: 1.8–3.0) MACE: HR 2.7; 95% CI: 1.7–4.5 Hospital admissions HR 3.4 (95% CI: 2.8–4.1) AECOPD hospital admissions: HR 5.2; 95% CI: 3.8–7.1 Community exacerbations: HR 2.1 (95% CI: 1.8–2.5) FI Mortality HR 2.6 (95% CI: 1.7–4.0) MACE HR 6.8 (95% CI: 2.7–17.0) Hospital admission HR 3.7 (95% CI: 2.5–5.4) AECOP hospital admissions HR 4.3; 95% CI: 2.4–7.7 Community exacerbations HR 2.4 (95% CI: 1.7–3.3)
Zhang, D. et al. [28]	Prospective study	302	1-year mortality risk FFP: HR = 3.11 (95% CI: 1.30–7.44) CFS: HR = 3.68 (95% CI: 1.03–13.16) SPPB: HR = 3.74 (95% CI: 1.39–10.06)
Gu, J. J. et al. [34]	Observational retrospective study	154	FI-lab increased AECOPD and mortality: OR 8.705 (95% CI: 3.646–20.782)
Bernabeu-Mora, R. et al. [38]	Prospective cohort	103 hospitalized COPD patients	Hospital readmission (45% vs. 18%) 90-day readmission (OR = 5.19; 95% CI: 1.26–21.50)
Chin, M. et al. [41]	Prospective study	46	Severe frailty vs. managing well and vulnerable: Total length of stay: 11 days [IQR 10–12] vs. 4 [IQR 2–7] Total cost CAD 14,109 [IQR 13,182–15,037] vs. 4366 [IQR 2490–7094] Previous hospitalization in the last 2 years, 6 [IQR 6–6] vs. 1 [IQR 0–2]
Patino-Hernandez, D. et al. [42]	Longitudinal study	2706 patients (76.4 years)	3-year mortality risk HR 1.95 (95% CI: 1.18–3.2)

Legend: COPD = chronic obstructive pulmonary disease; OR = odds ratio; HR = hazard ratio; AECOPD = acute exacerbation chronic obstructive pulmonary disease; IRR = incidence rate ratio; IQR = interquartile rank; CAD = Canadian dollars; MACE = major adverse cardiovascular events; FFP = Fried frailty phenotype; FI = frailty index; CFS = clinical frailty scale; SPPB = short physical performance battery.

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
