# Peer review of "Is Frailty Diagnosis Important in Patients with COPD? A Narrative Review of the Literature"

_ijerph, 2023, doi:10.3390/ijerph20031678_

Round 1
Reviewer 1 Report
Dear Authors,
the manuscript is intriguing, given that COPD is a burdensome and complex condition, in which comprehensive and multidisciplinary tailored interventions play a crucial role in patients’ outcomes.
However, in my opinion, some issues should be addressed to make the paper suitable for publication.
Major reviews
INTRODUCTION: This Section should be improved highlighting the need for early identification of physical frailty in order to develop individualized therapeutic interventions.
In accordance, you should cite the following references:
- Lippi L, Folli A, Curci C, D'Abrosca F, Moalli S, Mezian K, de Sire A, Invernizzi M. Osteosarcopenia in Patients with Chronic Obstructive Pulmonary Diseases: Which Pathophysiologic Implications for Rehabilitation? Int J Environ Res Public Health. 2022 Nov 2;19(21):14314. doi: 10.3390/ijerph192114314.
- de Sire A, Lippi L, Aprile V, Calafiore D, Folli A, D'Abrosca F, Moalli S, Lucchi M, Ammendolia A, Invernizzi M. Pharmacological, Nutritional, and Rehabilitative Interventions to Improve the Complex Management of Osteoporosis in Patients with Chronic Obstructive Pulmonary Disease: A Narrative Review. J Pers Med. 2022 Oct 1;12(10):1626. doi: 10.3390/jpm12101626.
- Vázquez-Gandullo E, Hidalgo-Molina A, Montoro-Ballesteros F, Morales-González M, Muñoz-Ramírez I, Arnedillo-Muñoz A. Inspiratory Muscle Training in Patients with Chronic Obstructive Pulmonary Disease (COPD) as Part of a Respiratory Rehabilitation Program Implementation of Mechanical Devices: A Systematic Review. Int J Environ Res Public Health. 2022 May 3;19(9):5564. doi: 10.3390/ijerph19095564.
MATERIALS AND METHODS: Concerning paper selection, this section should be implemented clarifying whether the duplication exclusion was performed manually or via automated tools. Moreover, were articles written in languages other than English assessed as well? This issue should be clarified in Results Section.
RESULTS. This section is too long and redundant with data presented in tables. I suggest better summarizing in tables some of the data presented in main text and avoiding repetitions.
DISCUSSION: My major concern is that the previous meta-analysis by Marengoni et al. in 2018 addressed the same topic as your article. In my opinion, your narrative review is necessary and it’s a good work. However, you should better emphasize the gap of knowledge you might overcome with your manuscript and why an up to date is necessary. Moreover, you should better highlight the differences between the meta-analysis by Marengoni et al. and clarify why did not address this topic in a systematic way.
DISCUSSION: Page 16, line 425. Indeed, you might consider the papers previously suggested, as many implications of medications in pathophysiology of COPD and frailty are addressed.
DISCUSSION: I think that the section is well-focused. However, it should be noted that databases searched were limited. This might have excluded some data. Thus, discussion should be implemented in light of this consideration.
Minor issues
INTRODUCTION: Page 1-2, lines 43-45. Please, review this period’s syntax
INTRODUCTION: Page 2, line 66. Please, provide average age of younger populations
RESULTS: Page 4, line 134; page 6, line 237. “A cross-sectional study conducted in 57 patients […]”. I think that other connectors rather than “in” would suite this period’s syntax. Please, rephrase this period accordingly.
RESULTS: Page 4, line 160. “frailty. Impairment”. Please, remove full stop.
RESULTS: Page 4, line 164. Please, define the acronym CGA.
RESULTS: Please, note that excessive use of acronyms should be avoided. There are some abbreviations which are not consistently used in the paper. I think that those following should be removed: NETT, CNS, HADS, ALC, ICARE. On the other hand, on page 7, line 264, “(FIND)” should be removed, as well, because its definition was already provided.
RESULTS: Page 6, first paragraph. Please, provide p values, whether available.
RESULTS: Page 6, line 213. Please, define IQR acronym.
RESULTS: Page 7, line 257. Please, correct the typo (i.e. “ÎQR”).
RESULTS: Page 9, line 349. Please, correct “are performed” with “were performed”.
RESULTS: Figures, Tables and Schemes. Please, note that each table, figure and scheme should be reported in the paper close to its first mention in the manuscript. Please, correct accordingly.
DISCUSSION: The first period’s syntax should be reviewed.
DISCUSSION: Page 15, line 422. Please, furtherly discuss the link between HIV/AIDS and COPD. Otherwise, remove this sentence.
Author Response
REVIEWER #1
The authors would like to thank reviewer #1 for the valuable comments to improve the quality of the manuscript. On behalf of my co-authors, we very much appreciate the time and effort you have put into your comments on our manuscript. Your reviews are most helpful. We have carefully reviewed the comments and thoroughly revised the manuscript accordingly. Our responses are given in a point-by-point manner below. We have submitted a revised version of our manuscript. All the changes are marked in yellow color.
Major reviews
INTRODUCTION: This Section should be improved highlighting the need for early identification of physical frailty in order to develop individualized therapeutic interventions.
In accordance, you should cite the following references:
- Lippi L, Folli A, Curci C, D'Abrosca F, Moalli S, Mezian K, de Sire A, Invernizzi M. Osteosarcopenia in Patients with Chronic Obstructive Pulmonary Diseases: Which Pathophysiologic Implications for Rehabilitation? Int J Environ Res Public Health. 2022 Nov 2;19(21):14314. doi: 10.3390/ijerph192114314.
- de Sire A, Lippi L, Aprile V, Calafiore D, Folli A, D'Abrosca F, Moalli S, Lucchi M, Ammendolia A, Invernizzi M. Pharmacological, Nutritional, and Rehabilitative Interventions to Improve the Complex Management of Osteoporosis in Patients with Chronic Obstructive Pulmonary Disease: A Narrative Review. J Pers Med. 2022 Oct 1;12(10):1626. doi: 10.3390/jpm12101626.
- Vázquez-Gandullo E, Hidalgo-Molina A, Montoro-Ballesteros F, Morales-González M, Muñoz-Ramírez I, Arnedillo-Muñoz A. Inspiratory Muscle Training in Patients with Chronic Obstructive Pulmonary Disease (COPD) as Part of a Respiratory Rehabilitation Program Implementation of Mechanical Devices: A Systematic Review. Int J Environ Res Public Health. 2022 May 3;19(9):5564. doi: 10.3390/ijerph19095564.
The authors agree with reviewer #1 on the need to emphasise the need for early identification of frailty in COPD patients in the introduction. Therefore, the authors have added the following text in this section:
Finally, it should not be forgotten that several common risk factors shared by COPD and frailty have been identified, including alcohol consumption, cigarette smoking, low physical activity, poor diet, and older age [15]. All of them are related to functional and cognitive decline. For these reasons, an early diagnosis of frailty is needed to provide a specific and individualised care plan to recover or mitigate the deleterious effects of frailty in these patients. Clinicians should actively screen frailty in COPD patients to improve their evolution. [15,16]
MATERIALS AND METHODS: Concerning paper selection, this section should be implemented clarifying whether the duplication exclusion was performed manually or via automated tools. Moreover, were articles written in languages other than English assessed as well? This issue should be clarified in Results Section.
The authors would like to thank reviewer #1 for the comment. The authors have added the following text to the manuscript:
The authors used an Excel® sheet to record all the bibliographic references in order to identify and exclude duplicated papers. The Excel sheet was also used to work with the selected articles and proceed to the selection of the articles. For this process, a bibliographic manager was used.
Regarding the second question, in the methods section the authors specified that the search was limited, in the aforementioned search engines, to English and Spanish languages. The authors have added a paragraph in the results section emphasising this point.
The articles were selected by four researchers based on the following eligibility criteria: the articles included in the criteria were meta-analyses, randomised clinical trials, cohort studies, case-control studies, observational studies and before-and-after studies; the population needed for the review were patients of 65 years old or over with an established COPD diagnosis; and prevalence and incidence studies involving reported frailty assessment tools and frailty prevalence were included along with interventions to reverse frailty. It was stated in the exclusion criteria that letters to the Editor, case reports, narrative and systematic reviews without meta-analyses, articles with no available abstract or those with only the abstract published, and studies that, despite meeting the inclusion criteria, less than 50% of their study sample was under the age of 65 (i.e. predominantly non-geriatric), did not comply with the aim of this review.
RESULTS. This section is too long and redundant with data presented in tables. I suggest better summarizing in tables some of the data presented in main text and avoiding repetitions.
The authors thank reviewer #1 for the comment. Following the suggestions of reviewer #1, the text has been reduced, avoiding repetitions and summarising the main results in the tables. The changes can be seen in the corresponding section of the manuscript.
DISCUSSION: My major concern is that the previous meta-analysis by Marengoni et al. in 2018 addressed the same topic as your article. In my opinion, your narrative review is necessary and it’s a good work. However, you should better emphasize the gap of knowledge you might overcome with your manuscript and why an up to date is necessary. Moreover, you should better highlight the differences between the meta-analysis by Marengoni et al. and clarify why did not address this topic in a systematic way.
The authors thank reviewer #1 for the valuable comment. In the discussion section the authors have added:
The authors of the aforementioned meta-analysis emphasised the common mechanisms and risk factors of COPD and frailty, and the increased risk of mortality and hospital readmissions for acute exacerbation of COPD in frail patients. However, this meta-analysis did not assess the relationship between frailty, geriatric syndromes and COPD, as reflected in this narrative review, nor did it report other types of adverse events associated to frail COPD patients, as described in this narrative review.
Likewise, in the limitations subsection, the authors have added:
The limitations of the review stem from the absence of randomised clinical trials and the considerable number of cross-sectional studies that allowed finding associations, although not causality. The authors evaluated the published articles included in the OVID and PubMed databases in order to include articles published in journals with the greatest impact. However, this decision is not exempt from bias, as publications included in other databases that could have been of interest for this narrative review have not been evaluated. Likewise, the authors are aware that a systematic review could have provided more evidence on the topics covered. However, the heterogeneity of the frailty assessment instruments, the different settings in which the patients were recruited, and the different severity degrees of the disease in the selected studies led the authors to dismiss this possibility. Given the growing interest in this topic, the authors of this review believe it is possible to develop a systematic review in a near future.
DISCUSSION: Page 16, line 425. Indeed, you might consider the papers previously suggested, as many implications of medications in pathophysiology of COPD and frailty are addressed.
The authors thank reviewer #1 for this consideration and have added the following text to the manuscript:
As a recent review described, beyond chronic inflammation and reduced physical activity, factors that decrease muscle strength and endurance in COPD patients include oxidative stress, inactivity, hypoxaemia, hormone abnormality, lack of nutrients such as protein and vitamin D, and the use of corticosteroids [15].
DISCUSSION: I think that the section is well-focused. However, it should be noted that databases searched were limited. This might have excluded some data. Thus, discussion should be implemented in light of this consideration.
The authors agree with reviewer #1 about the bias of having evaluated only the papers collected by the OVID and PubMED databases. For this reason, the authors have included the following text in the ‘limitations’ subsection:
The authors evaluated the published articles included in the OVID and PubMed databases in order to include articles published in journals with the greatest impact. However, this decision is not exempt from bias, as publications included in other databases that could have been of interest for this narrative review have not been evaluated.
Minor issues
INTRODUCTION: Page 1-2, lines 43-45. Please, review this period’s syntax.
The authors thank reviewer #1 for the comment. The period’s syntax was changed to “Frailty is dynamic in nature, and it is possible to find positive and negative trajectories among the robust, pre-frailty and frailty categories, both spontaneously and after interventions [3].”
INTRODUCTION: Page 2, line 66. Please, provide average age of younger populations.
The authors thank the reviewer for the comment. The sentence between parentheses has been added in the manuscript: under 65 years old.
RESULTS: Page 4, line 134; page 6, line 237. “A cross-sectional study conducted in 57 patients […]”. I think that other connectors rather than “in” would suite this period’s syntax. Please, rephrase this period accordingly.
The authors would like to thank reviewer #1 for the comment. The paragraph was rephrased as follows:
A cross-sectional study of 57 patients with COPD (GOLD 1-4) found the highest prevalence of frailty (EFIP) detected in this narrative review, an 83%, which was associated to malnutrition and physical frailty [40].
RESULTS: Page 4, line 160. “frailty. Impairment”. Please, remove full stop.
This section has been rewritten and this sentence has been removed.
RESULTS: Page 4, line 164. Please, define the acronym CGA.
The authors have defined the acronym CGA.
RESULTS: Please, note that excessive use of acronyms should be avoided. There are some abbreviations which are not consistently used in the paper. I think that those following should be removed: NETT, CNS, HADS, ALC, ICARE. On the other hand, on page 7, line 264, “(FIND)” should be removed, as well, because its definition was already provided.
The authors agree with the reviewer #1. The acronyms mentioned have been removed. Also, FiND definition was removed on page 7, line 264.
RESULTS: Page 6, first paragraph. Please, provide p values, whether available.
The authors added the p value.
RESULTS: Page 6, line 213. Please, define IQR acronym.
The authors have defined IQR acronym.
RESULTS: Page 7, line 257. Please, correct the typo (i.e. “ÎQR”).
The authors would like to thank reviewer #1 for the comment. The typo has been removed from the text.
RESULTS: Page 9, line 349. Please, correct “are performed” with “were performed”.
This section has been rewritten and this sentence has been removed.
RESULTS: Figures, Tables and Schemes. Please, note that each table, figure and scheme should be reported in the paper close to its first mention in the manuscript. Please, correct accordingly.
The authors thank reviewer #1 for the valuable comment. Figures, tables and diagrams have been reported in the paper closet or its first mention in the manuscript.
DISCUSSION: The first period’s syntax should be reviewed.
The authors would like to thank the reviewer for the comment. The period’s syntax was revised and modified to “In the present review, a great variability of the frailty measurement tools used was observed , as well as the association between the presence of frailty, a high prevalence of geriatric syndromes and COPD-related aspects, the close relationship between the presence of frailty and poorer health outcomes and, finally, a single study that explored the potential benefits of a combined intervention (pharmacological and non-pharmacological, including rehabilitation and a nutritional approach), showing improvements in functional parameters.”
DISCUSSION: Page 15, line 422. Please, furtherly discuss the link between HIV/AIDS and COPD. Otherwise, remove this sentence.
The authors thank reviewer #1 for the comment and have added the following text to the manuscript:
Just as the progression of diseases such as HIV/AIDS and higher associated inflammatory activity increases frailty prevalence compared to patients of the same age who do not suffer from these diseases [46], the prevalence of frailty in COPD patients is also higher, probably also associated with higher inflammatory activity.

Reviewer 2 Report
This study performed systematic review of the literature about the association between COPD and frailty. Although it is an important issue, there are some points to be revised.
1) Because a meta-analysis is included in this study, the authors should explain more precisely what kind of reviews were not excluded.
2) As a systematic review, the authors should be focused on the predetermined questions, i.e. frailty prevalence, frailty as a predictive factor of bad outcomes, and interventions on frailty COPD patients.
The present main document of Result 3.1 comprised of simple summaries of the articles, including the findings which are not related with the predetermined questions. The Result 3.1 should be revised in large part.
Meanwhile, the authors should make a section focused on the prevalence of geriatric syndromes, if this issue is also a predetermined question of this systematic review.
3) The authors should indicate what is novel compared to the previous systematic review (11).
Author Response
REVIEWER #2
The authors would like to thank reviewer #2 for the valuable comments to improve the quality of the manuscript. On behalf of my co-authors, we very much appreciate the time and effort you have put into your comments on our manuscript. Your reviews are most helpful. We have carefully reviewed the comments and thoroughly revised the manuscript accordingly. Our responses are given in a point-by-point manner below. We have submitted a revised version of our manuscript. All the changes are marked in yellow color.
1) Because a meta-analysis is included in this study, the authors should explain more precisely what kind of reviews were not excluded.
The authors thank reviewer #2 for the comment. The authors have added meta-analyses in the inclusion criteria and narrative and systematic reviews as exclusion criteria in the manuscript. The final version is shown in the manuscript as follows:
The articles were selected by four researchers based on the following eligibility criteria: the articles included in the criteria were meta-analyses, randomised clinical trials, cohort studies, case-control studies, observational studies and before-and-after studies; the population needed for the review were patients of 65 years old or over with an established COPD diagnosis; and prevalence and incidence studies involving reported frailty assessment tools and frailty prevalence were included along with interventions to reverse frailty. It was stated in the exclusion criteria that letters to the Editor, case reports, narrative and systematic reviews without meta-analyses, articles with no available abstract or those with only the abstract published, and studies that, despite meeting the inclusion criteria, less than 50% of their study sample was under the age of 65 (i.e. predominantly non-geriatric), did not comply with the aim of this review.
2) As a systematic review, the authors should be focused on the predetermined questions, i.e. frailty prevalence, frailty as a predictive factor of bad outcomes, and interventions on frailty COPD patients.
The authors of this narrative review thank reviewer #2 for the valuable comment. The authors have rewritten the results section to focus the narrative on the answer to the predetermined questions. Also to clarify the default questions, the authors have added this text at the end of the Introduction:
Given the increase of scientific literature on this topic, the authors considered important to perform a narrative review of the literature in order to find out the actual prevalence of frailty in patients with COPD, the association of frailty with other geriatric syndromes in these patients, the influence of frailty on their clinical evolution and survival and the therapeutic approaches to try to reverse or reduce frailty in COPD patients.
The present main document of Result 3.1 comprised of simple summaries of the articles, including the findings which are not related with the predetermined questions. The Result 3.1 should be revised in large part.
Following the recommendations of reviewer #2, the authors have rewritten the Results 3.1 section.
Meanwhile, the authors should make a section focused on the prevalence of geriatric syndromes, if this issue is also a predetermined question of this systematic review.
Following the advice and considerations of reviewer #2, in line with the previous comment, the authors have rewritten the section and added a subsection 3.1.1. titled Geriatric syndromes prevalence linked to frailty in COPD patients:
3.1.1. Geriatric syndromes prevalence linked to frailty in COPD patients
The frailty prevalence in COPD patients is linked to a high prevalence of geriatric syndromes, as four of the studies included in this narrative review showed. Thus, malnutrition and frailty coexisted in 40% of the COPD patients, malnutrition and physical frailty coexisted in 18% and malnutrition and disability coexisted in 21% of the cases in a cross-sectional study [40]. This overlap of geriatric patients should deliver a nutritional intervention in COPD patients, especially before starting a rehabilitation programme [40]. The EFIP and PG-SGA scores were significantly correlated (r=0.43, P=0.001), as well as Fried’s criteria and the PG-SGA score (r=0.37, P=0.005). A nutritional intervention should be delivered by health care professionals in COPD patients before starting a rehabilitation programme [40]. A case-control study [30] assessed the prevalence of geriatric syndromes in 150 COPD patients. Frailty was linked to a high prevalence of geriatric syndromes, such as impairment in Instrumental Activities of Daily Living (IADL), 37.3%; cognitive impairment, 35.3%; urinary incontinence, 20.7%; and malnutrition (20.7%). Two variables increased the risk of geriatric syndromes prevalence: dyspnoea (≥2 mMRC grade) and low socioeconomic status [30]. The authors emphasised the importance of the performance of the Comprehensive Geriatric Assessment (CGA) in frail COPD patients. Frailty has been associated to mood disorders in COPD patients. A cross-sectional study of 40 COPD patients reported a low perception of quality of life. In addition, patients classified as frail, using the KCL score, presented lower left and right hippocampal, subiculum and presubiculum volume compared to non-frail patients. The authors concluded by pointing at the impact of frailty on the hippocampal volume and its combined associations with poor quality of life [35]. Finally, a cross-sectional study conducted in 3222 older adults (69.6 [SD 7.4] years) with COPD that lived in the community analysed the prevalence of geriatric syndromes in the recruited sample, including frailty, functional disability, physical function impairment, low physical activity, falls, polypharmacy, loneliness, depression, cognitive impairment, urinary incontinence and comorbidity [37]. The prevalence of geriatric syndromes was higher in COPD patients than in non-COPD patients: 16% of frailty; 53.9% of urinary incontinence; 57.7% of loneliness perception; 58.1% of functional disability; 12.9% of moderate cognitive impairment; 32% of depressive symptoms; and 37.5% of severe polypharmacy (>10 prescribed drugs). Compared to non-COPD patients, the frailty adjusted OR was 6.3, 95%CI 3.0, 13.0. This means that COPD patients had a risk of frailty more than six times higher than controls. Likewise, the adjusted OR for functional disability was 1.4 95%CI 1.01, 2.0; for impaired physical function the adjusted OR was 2.1, 95%CI 1.1, 3.7; for extreme low physical activity the adjusted OR was 2.3, 95%CI 1.5, 3.5; for polypharmacy (≥10 medications) the adjusted OR was 2.9, 95%CI 2.0, 4.2; and for depression the adjusted OR was 1.9, 95%CI 1.4, 2.7) [37].
3) The authors should indicate what is novel compared to the previous systematic review (11).
The authors thank reviewer #2 for the valuable comment. Since the previous systematic review (17 in the new version) focused on the shared mechanisms between COPD and frailty and the increased risk of mortality from hospital readmissions for acute exacerbations of COPD, the authors of this narrative review have added this text in the discussion:
The authors of the aforementioned meta-analysis emphasised the common mechanisms and risk factors of COPD and frailty and the increased risk of mortality and hospital readmissions for acute exacerbation of COPD in frail patients. However, this meta-analysis did not assess the relationship between frailty, geriatric syndromes and COPD, as reflected in this narrative review, nor did it report other types of adverse events associated to frail COPD patients, as described in this narrative review.

Reviewer 3 Report
Frailty is relatively common in subjects with lung disease and appears to be associated with poor functional status, exacerbation of lung disease, disability, poor health, and reduced quality of life, even mortality. Assessing frailty can help clinicians identify patients at increased risk for poor outcomes, as well as suggest therapies to preserve functional independence, reduce disability, and improve survival.
The abstract should illustrate the objectives of this review a little more clearly. As for the title, it is informative enough to to draw its potential reader's attention.
The introduction of this article successfully explains why the current narrative review of the literature is important and the clinical implications of not screening for frailty in patients with COPD.
Here are some suggestions that would improve the manuscript:
1. It would be good in the introduction to give a short information about COPD as a disease, when frailty is most often manifested as a symptom (before, after diagnosis or could be due to a shared physiopathology)?
2. The explanation for the selection of the 25 articles used from the available 575 is quite extensive but it does not become very clear leading was to have all available information about a clinical trial, the age of patients with COPD, the measurement instruments, prevalence, criteria for frailty etc.? Which takes priority in selection?
3. If the authors follow the concept of the article, they could represent the data of representative instruments of measurement of frailty in use in table (frailty method, populations, scoring, etc.). (line 115-123).
4. In chapter 3.1. Frailty prevalence, the data from the studies are described and given below in a table 1. It is good to mention that table in the chapter. Depending on the assessment methods and heterogeneity of the population, the prevalence of frailty varies greatly.The general trend of frailty prevalence increases with age. The prevalence of frailty in patients with COPD has been reported to range from 6% to 85,9%. Since the data in the studies were measured with different frailty instruments, some of them depending on the populations, severity of COPD and accompanying diseases it appears that no exact specifics can be drawn. How comparable are these results then? Shouldn't the studies be categorized by a certain criteria (e.g. instrument and severity) and considered in groups? (this is just a suggestion).
5. Line 327, 332 – references?
6. In chapter 3.2. Frailty as a predictive factor of bad outcomes, showed that in 12 articles frailty was used as a predictive factor, but frailty was also associated with a risk of hospitalizations in patients without COPD (line 311, reference 25). Could frailty really be used as an indicator of an upcoming worsening condition in patients with COPD or other lung disease in a real clinical practice if patients are screened periodically after diagnosis?
7. In chapter 3.3.Interventions on frailty COPD patients, additional references related to therapeutic, palliative and rehabilitative interventions can be added.
8. In chapter “Discussion” describes very well the importance of the problem, the limitations of the manuscript as well as some possible solutions. To highlight and illustrate the the importance of frailty diagnosis inpatients with COPD, a general scheme/figure can be given in associations with pathobiological perturbations (chronic inflammation, cognitive impairment, endocrine dysregulation, sarcopenia etc), and hypothesized genetic, molecular, and functional changes underlying the frailty phenotype (immune senescence, mitochondrial dysfunction, telomere attrition, genome instability, epigenetic alteration, etc.).
9. Can the authors speculate whether the diagnosis would be more accurate if a particular combination of frailty measurement instruments were applied? If there were an established and/or standard protocol for measuring frailty on admission to the hospital, would this help for the status and treatment of patients?
The article is written very detailed and the tables correctly illustrate the information. The reader understands the importance of the topic although the large volume of scientific data. Providing readers with possible hypotheses and solutions in the context of scientific facts would also be helpful.
All these refinements may make the article more interesting for readers from multiple backgrounds.
Author Response
REVIEWER #3
The authors would like to thank reviewer #3 for the valuable comments to improve the quality of the manuscript. On behalf of my co-authors, we very much appreciate the time and effort you have put into your comments on our manuscript. Your reviews are most helpful. We have carefully reviewed the comments and thoroughly revised the manuscript accordingly. Our responses are given in a point-by-point manner below. We have submitted a revised version of our manuscript. All the changes are marked in yellow color.
Frailty is relatively common in subjects with lung disease and appears to be associated with poor functional status, exacerbation of lung disease, disability, poor health, and reduced quality of life, even mortality. Assessing frailty can help clinicians identify patients at increased risk for poor outcomes, as well as suggest therapies to preserve functional independence, reduce disability, and improve survival.
The abstract should illustrate the objectives of this review a little more clearly. As for the title, it is informative enough to to draw its potential reader's attention.
The authors thank reviewer #3 for the comment. The objectives of the review have been clarified in the abstract following the recommendations of reviewer #3. The following text has been added to the manuscript:
…was carried out to address three questions: frailty and other geriatric syndromes prevalence in COPD patients, the link between frailty and worse health outcomes in COPD patients and the non-pharmacological interventions performed in order to reverse frailty in these patients.
The introduction of this article successfully explains why the current narrative review of the literature is important and the clinical implications of not screening for frailty in patients with COPD.
The authors thank reviewer #3 for the complimentary comment on the introduction.
Here are some suggestions that would improve the manuscript:
- It would be good in the introduction to give a short information about COPD as a disease, when frailty is most often manifested as a symptom (before, after diagnosis or could be due to a shared physiopathology)
The authors thank reviewer #3 for the comment and have added the following text in the introduction:
Likewise, COPD is the third leading cause of death worldwide. Many factors contribute to the development of COPD, including genetic factors (alpha1-antitrypsin deficiency), pollution, cigarette smoking, and occupational exposure to various chemicals. COPD manifests as an inflammatory disease that affects the airways, lung parenchyma and pulmonary vasculature. Inhalation exposure can trigger an inflammatory response, leading to a decrease in forced expiratory volume and tissue destruction, leading to airflow limitation. Airflow limitation is the main pathophysiologic feature of COPD [11]. This relationship between both entities may be because COPD and frailty share some risk factors, including ageing, smoking, inflammation [12], and clinical manifestations such as fatigue, anorexia, muscle weakness, and slow walking speed [13]. In this sense, Lahousse et al. suggested that both pathologies might have a common physiopathology [14].
Finally, it should not be forgotten that several common risk factors shared by COPD and frailty have been identified, including alcohol consumption, cigarette smoking, low physical activity, poor diet, and older age [15]. All of them are related to functional and cognitive decline. For these reasons, an early diagnosis of frailty is needed to provide a specific and individualised care plan to recover or mitigate the deleterious effects of frailty in these patients. Clinicians should actively screen frailty in COPD patients to improve their evolution. [15,16]
- The explanation for the selection of the 25 articles used from the available 575 is quite extensive but it does not become very clear leading was to have all available information about a clinical trial, the age of patients with COPD, the measurement instruments, prevalence, criteria for frailty etc.? Which takes priority in selection?
The authors thank reviewer #3 for the appropriate comment. The authors have added the following text in the methods section:
The articles were selected by four researchers based on the following eligibility criteria: the articles included in the criteria were meta-analyses, randomised clinical trials, cohort studies, case-control studies, observational studies and before-and-after studies; the population needed for the review were patients of 65 years old or over with an established COPD diagnosis; and prevalence and incidence studies involving reported frailty assessment tools and frailty prevalence were included along with interventions to reverse frailty. It was stated in the exclusion criteria that letters to the Editor, case reports, narrative and systematic reviews without meta-analyses, articles with no available abstract or those with only the abstract published, and studies that, despite meeting the inclusion criteria, less than 50% of their study sample was under the age of 65 (i.e. predominantly non-geriatric), did not comply with the aim of this review.
- If the authors follow the concept of the article, they could represent the data of representative instruments of measurement of frailty in use in table (frailty method, populations, scoring, etc.). (line 115-123).
The authors thank reviewer #3 for the comment. The suggested data have been included in Table 1. In addition, the number of studies included in the review that used each instrument has been specified in the manuscript. The following text has been added to the manuscript:
There was a considerable variation regarding frailty measurement tools, as up to 13 tools used in the studies included in the review were found: Hospital Frailty Risk Score (HFRS) in two studies; Fried Frailty Phenotype (FFP) in eight studies; Frailty Index (FI) in four studies; Kihon Checklist (KCL) in two studies; The Fatigue, Resistance, Ambulation, Illnesses and Loss of weight (FRAIL) in two studies; Reported Edmonton Frail Scale (REFS), Chinese-Canadian Study of Health and Aging Clinical Frailty Scale, Evaluative Frailty Index for Physical Activity (EFIP), Clinical Frailty Scale (CFS), Short Physical Performance Test (SPPB), The Frail Non-Disabled questionnaire (FiND), Timed Up and Go (TUG) and FI-lab in one study each.
- In chapter 3.1. Frailty prevalence, the data from the studies are described and given below in a table 1. It is good to mention that table in the chapter. Depending on the assessment methods and heterogeneity of the population, the prevalence of frailty varies greatly. The general trend of frailty prevalence increases with age. The prevalence of frailty in patients with COPD has been reported to range from 6% to 85,9%. Since the data in the studies were measured with different frailty instruments, some of them depending on the populations, severity of COPD and accompanying diseases it appears that no exact specifics can be drawn. How comparable are these results then? Shouldn't the studies be categorized by a certain criteria (e.g. instrument and severity) and considered in groups? (this is just a suggestion).
The authors thank reviewer #3 for the suggestion. It is true, as commented in the discussion, that the different frailty measurement instruments evaluate different domains and are heterogeneous. However, the two articles that compare the different measurement instruments obtain similar frailty prevalence and prognostic abilities. Likewise, regardless of the severity and the instrument, the prevalence of frailty is uneven, regardless of the stability of the COPD, the age range of the sample and the instrument). However, the prognostic capacity of the presence of frailty in patients with COPD persists despite this heterogeneity of ages, instruments, and degrees of severity of COPD, confirming the importance of the presence of frailty in the evolutionary course of these patients. For this reason, the authors have added the following text in the discussion section:
Therefore, the prevalence of frailty was lower when using the FFP (17%) than the FI [28] although when using four tools (FFP, 51%; SPPB, 58.6%; FI-CD 59.6%; CFS, 64.2%), the authors reported a moderate-to-substantial agreement between the instruments [29]. Despite the concordance reported in the previous study [30], the heterogeneity of the assessed domains in the different frailty tools [4] and the severity of COPD, the prevalence of frailty is uneven in the different studies included in this review, regardless of the stability of the COPD, the setting where the patients were recruited, the age range of the sample, or the instrument (even within the same instrument). Nevertheless, the prognostic capacity of the presence of frailty in patients with COPD persists despite this heterogeneity of ages, instruments, and degrees of severity of COPD, confirming the importance of the presence of frailty in the evolutionary course of these patients.
- Line 327, 332 – references?
The authors have added the corresponding reference [30] in the text, although it has been changed due to the suggestion of reviewer #1, who requested that the section be summarized.
- In chapter 3.2. Frailty as a predictive factor of bad outcomes, showed that in 12 articles frailty was used as a predictive factor, but frailty was also associated with a risk of hospitalizations in patients without COPD (line 311, reference 25). Could frailty really be used as an indicator of an upcoming worsening condition in patients with COPD or other lung disease in a real clinical practice if patients are screened periodically after diagnosis?
The authors thank reviewer #3 for the pertinent comment. Frailty is a predictor of worse outcomes in patients with and without COPD. References 50 and 51 could serve as an example. It is true, as reviewer #3 comments, in line with what has been said, that the relationship between frailty and poorer health outcomes is not specific for patients with COPD. However, based on the results of the review, patients with COPD who were also frail had worse health outcomes than COPD patients without frailty. Therefore, frailty is indeed an indicator of worse evolution in COPD patients and should be detected early in real life to try to reverse it and avoid this clinical worsening in the evolutionary course. The authors have added the following text in the discussion:
As previously mentioned, frailty is a predictor of worse health outcomes [50] and mortality [51] in older adults. Thus, it is also in older adults with COPD [20,22-25,27-29, 34,38,41,43]. In fact, COPD patients who were also frail had worse outcomes than COPD patients without frailty. Therefore, we can consider that frailty is an indicator of worse evolution in COPD patients and should be detected early in real life thought a CGA to try to reverse it and avoid this worsening in the clinical course.
- In chapter 3.3.Interventions on frailty COPD patients, additional references related to therapeutic, palliative and rehabilitative interventions can be added.
The authors agree with reviewer #3 that palliative and rehabilitative interventions could be discussed in more detail, but since no more articles were found in the search for the narrative review, the authors have added the references in the discussion and not in the results section. The following text has been added to the manuscript:
Home-based pulmonary rehabilitation may also be another way to treat frailty in these patients, which could improve functional capacity, frailty status, quality of life and pulmonary symptoms such as fatigue [55]. Palliative care should not be forgotten. As a recent review recalls, palliative care is much more than palliative care or end-of-life care [56] and could be integrated with the above-mentioned interventions and approaches to provide holistic and comprehensive assessment and treatment, pharmacological and non-pharmacological, to improve symptom control and quality of care for these patients.
- In chapter “Discussion” describes very well the importance of the problem, the limitations of the manuscript as well as some possible solutions. To highlight and illustrate the the importance of frailty diagnosis inpatients with COPD, a general scheme/figure can be given in associations with pathobiological perturbations (chronic inflammation, cognitive impairment, endocrine dysregulation, sarcopenia etc), and hypothesized genetic, molecular, and functional changes underlying the frailty phenotype (immune senescence, mitochondrial dysfunction, telomere attrition, genome instability, epigenetic alteration, etc.).
The authors thank reviewer #3 for their suggestion and have included a figure summarising the link between Pathophysiological disorders and factors associated to frailty phenotype. Figure 2 has been added to the manuscript.
- Can the authors speculate whether the diagnosis would be more accurate if a particular combination of frailty measurement instruments were applied? If there were an established and/or standard protocol for measuring frailty on admission to the hospital, would this help for the status and treatment of patients?
The article is written very detailed and the tables correctly illustrate the information. The reader understands the importance of the topic although the large volume of scientific data. Providing readers with possible hypotheses and solutions in the context of scientific facts would also be helpful.
All these refinements may make the article more interesting for readers from multiple backgrounds.
The authors thank reviewer #3 for the valuable comment and have added the following text to the discussion section:
Certainly, the authors have not been able to determine through this narrative review which frailty tool is the most indicated for patients with COPD, or whether it is necessary to use several scales altogether (nor could the authors advise on the appropriate combination of instruments) to improve the sensitivity and specificity of the diagnosis of frailty. Once again, given the importance of frailty in the clinical course of COPD patients, the authors considered the need to emphasise the protocolisation of frailty screening in both outpatient and inpatient settings. It could improve (prior inclusion of the patient in the appropriate interventions) or reverse frailty and other associated geriatric syndromes, improve the evolution of the disease (reducing AECOPD, hospital admissions and mortality), and maintain or improve the quality of life of COPD patients. Though the discussion is the appropriate section for speculation leading to new hypotheses, such hypotheses need to be ratified by relevant studies confirming them, at best, or refuting them, at worst.

Round 2
Reviewer 1 Report
Dear Authors,
in my opinion, the manuscript is interesting. You have significantly improved the paper during the revision process.
Therefore, in my opinion, the paper is now suitable for publication in this Journal.
Best regards
Reviewer 2 Report
The manuscript was revised well.
Reviewer 3 Report
Dear Authors,
I have been carefully reviewed your revised article. In my opinion, this revised article incorporates all of the points raised in the original draft to the best of my knowledge. The work is a significant contribution to the field and sounds scientifically.Best wishes to all of the authors who contributed to the production of this wonderful work and congratulations on their future endeavors.